# Gastric Carcinomas with Stromal B7-H3 Expression Have Lower Intratumoural CD8+ T Cell Density

**DOI:** 10.3390/ijms22042129

**Published:** 2021-02-21

**Authors:** Dita Ulase, Hans-Michael Behrens, Sandra Krüger, Sebastian Zeissig, Christoph Röcken

**Affiliations:** 1Department of Pathology, Christian-Albrechts-University, 24105 Kiel, Germany; dita.ulase@rsu.lv (D.U.); behrensm@path.uni-kiel.de (H.-M.B.); Sandra.Krueger@uksh.de (S.K.); 2Department of Pathology, Riga Stradins University, LV-1007 Riga, Latvia; 3Department of Medicine I, University Medical Center Dresden, Technische Universität (TU) Dresden, 01307 Dresden, Germany; sebastian.zeissig@tu-dresden.de; 4Center for Regenerative Therapies Dresden (CRTD), Technische Universität (TU) Dresden, 01307 Dresden, Germany

**Keywords:** B7-H3, CD8-Positive T-Lymphocytes, gastric cancer, immune checkpoint, immune evasion, immunohistochemistry, tumour escape, tumour microenvironment

## Abstract

CD8+ T cells are the main effector cells of anti-cancer immune response that can be regulated by various costimulatory and coinhibitory molecules, including members of the B7 family. B7 homolog 3 (B7-H3) appears as a promising marker for immunotherapy; however, its significance in gastric cancer (GC) is unclear yet. We evaluated the spatial distribution of CD8+ T cells in relation to the expression of B7-H3 by double immunohistochemical staining. The level of B7-H3 intensity was scored manually (0–3) and dichotomized into B7-H3-low and B7-H3-high groups. The distribution and density of CD8+ T cells was analysed using whole slide digital imaging. B7-H3 was expressed mainly in the stromal compartment of GC (*n* = 73, 76% of all cases). Tumours with high expression of B7-H3 showed larger spatial differences of CD8+ T cells (86.4/mm^2^ in tumour centre vs. 414.9/mm^2^ in invasive front) when compared to B7-H3-low group (157.7/mm^2^ vs. 218.7/mm^2^, respectively) (*p* < 0.001). This study provides insight into the expression pattern of B7-H3 in GC of Western origin. In GCs with higher level of B7-H3 expression, CD8+ T cells were spatially suppressed in the tumour centre suggesting that B7-H3 might be involved in tumour escape mechanisms from the immune response.

## 1. Introduction

The tumour immune microenvironment (TIME) can be categorized into three classes [1]: (a) the infiltrated-excluded TIME that is characterized by the exclusion of cytotoxic T cells (CTL) from the tumour core; (b) the infiltrated-inflamed TIME that is observed in immunologically “hot” tumours and characterized by high infiltration of CTLs expressing programmed cell death protein 1 (PD-1) and leukocytes and tumour cells expressing the immune-dampening PD-1 ligand PD-L1; and (c) a subclass of infiltrated-inflamed TIME, i.e., TLS-TIME, that includes tertiary lymphoid structures and lymphoid aggregates whose cellular composition is similar to that found in lymph nodes [1]. TIME is a function of both tumour genotype/phenotype and immunological composition. CD8+ CTLs are the main effector cells of the adaptive immune system. They are able to recognize tumour-specific antigens and destroy cancer cells by perforin and granzyme-mediated apoptosis. Studies show that a higher amount of CD8+ T cells in the tumour is associated with better prognosis in gastric cancer (GC) [2,3]. However, during cancer progression they become inactivated due to the immunosuppressive nature of TIME. Moreover, lack of T cell infiltration is one of the major factors involved in the resistance to immune checkpoint inhibitors [4].

B7 homolog 3 protein (B7-H3, or CD276) is one of the immune checkpoint molecules of the B7 family. Although the exact receptor of B7-H3 has not been discovered yet [5], this molecule is overexpressed in various cancers and involved in immune evasion [5,6]. One of the theories include inhibition of CD8+ T cells [7]. B7-H3 is found both in tumour cells and its TIME, including fibroblasts, endothelial cells, natural killer cells, B-cells, macrophages, and dendritic cells [8]. Moreover, it seems to be linked to cancer progression, metastatic potential and worse prognosis in several malignancies including non-small cell lung cancer, breast cancer, prostate cancer, renal cell cancer, and colorectal cancer [6]. To our knowledge, only one Asian cohort study has assessed B7-H3 expression in relation to the distribution of CD8+ T cells in GC [9]. There are currently no data about B7-H3 expression in GC of Western countries. To better understand this complex interaction, we analysed the expression of B7-H3 and the spatial distribution of CD8+ T cells on whole tissue sections by using double immunohistochemical staining.

## 2. Results

### 2.1. B7-H3 Expression and Its Association with Clinicopathological Characteristics

Clinicopathological parameters of the cohort are summarized in the Table 1. A total of 96 cases (intestinal, *n* = 48; diffuse, *n* = 27; mixed, *n* = 6; unclassified, *n* = 15) were included in the study. Among 73 B7-H3 positive GCs (76%), B7-H3 expression was found within the tumour stroma (*n* = 60) or both in stromal cells and neoplastic cells (*n* = 13). After assessing the intensity of B7-H3 expression, 55 cases showed negative or low staining, and 41 cases showed moderate or strong staining (Figure 1A–C). Since the expression levels of B7-H3 in tumour cells were low, the remainder of the study focused on the stromal expression.

The associations between B7-H3 staining intensity and clinicopathological parameters are detailed in Table 1. In this cohort, the B7-H3 stromal expression correlated with tumour location, Lauren phenotype, and pT category. No significant associations were found between B7-H3 expression and patients’ gender or age, tumour histological grade, presence of nodal or distant metastases, Union for International Cancer Control (UICC) stage, and status of resection lines.

### 2.2. Survival Analysis

The median overall survival (OS) for the entire cohort was 17.5 months, and the median cause-specific survival (CSS) was 18.4 months. No significant differences were found in OS and CSS between B7-H3-low and B7-H3-high groups (*p* = 0.784 and *p* = 0.990, respectively).

### 2.3. CD8+ T Cell Infiltration in Relation to B7-H3 Expression

The median density of CD8+ T cells (cells/mm^2^) differed significantly between both tumour compartments (*p* < 0.001; Table 2). In general, the median number of CD8+ T cells was lower in the tumour centre (TC) (112.5 cells/mm^2^) than in the invasive front (IF) (258.1 cells/mm^2^). Tumours with higher B7-H3 expression in the stroma showed greater spatial differences of CD8+ T cells (86.4/mm^2^ in TC and 414.9/mm^2^ in IF) when compared to the B7-H3-low group (157.7/mm^2^ and 218.7/mm^2^, respectively; *p* < 0.001) (Figure 1D–F, Figure 2, and Table 2). Additionally, a specific pattern was observed: the more intense the stromal B7-H3 staining was, the lower the CD8+ T cell density in TC and, at the same time, CD8+ T cells were concentrated in the IF.

### 2.4. TCGA Dataset Analysis

Data from The Cancer Genome Atlas Stomach Adenocarcinoma (TCGA STAD) cohort were retrieved for quantitative validation analysis using TCGA Xena computational tool (UC Santa Cruz) [10]. The B7-H3 mRNA expression in GC (TCGA STAD, *n* = 415) was significantly higher than that in normal gastric tissue (TCGA STAD, *n* = 35) (*p* < 0.001, Wilcoxon test, Appendix A). The effect of B7-H3 expression on CD8+ T cell infiltration and survival data in TCGA cohort was analysed by using TIMER2.0 web platform [11]. In TCGA STAD cohort, the expression of B7-H3 was significantly and negatively correlated with CD8+ T cell infiltration (r = −0.224, *p* < 0.001, Appendix A). Similar to our cohort, no statistically significant difference was found in OS between B7-H3 expression groups in the validation dataset (HR = 1.03, *p* = 0.661).

## 3. Discussion

Studies have demonstrated that the analysis of tumour infiltrating lymphocytes (TIL) alone cannot explain the complex crosstalk between tumour cells and their TIME. Here, we have presented a potential relationship between B7-H3, an immune checkpoint molecule, and CD8+ T cells, the main effector cells involved in tumour surveillance. The major findings of this study are the following: (1) B7-H3 is expressed mainly within the tumour stroma; (2) B7-H3 expression correlates significantly with tumour location, Lauren phenotype, and pT category; (3) median density of CD8+ T cells differs significantly between TC and IF; (4) in cases with higher stromal B7-H3 expression, restricted CD8+ cell infiltration in the TC was observed. Taken together, our results suggest a potential B7-H3 role in the immunosuppressive TIME of GC. However, while there is no specific human receptor for B7-H3 described yet, the exact mechanism of this interaction remains unclear.

Most studies of TILs (including CD8+ T cells) and B7-H3 in GC are carried out on Asian cohorts. According to their meta-analyses, higher intratumour CD8 expression is associated with an improved overall survival [2,12]. Distinct tumour immunity signatures between Asian and non-Asian GCs have been found, including higher expression of CD8 in Western GCs [13]. It is postulated that intratumoural CD8+ T cells are not randomly distributed but accumulate in an organized manner [14]. Different histological types may differ in their ability to recruit CD8+ T cells into the TC and IF. Moreover, tumours with higher TIL infiltrates have better adaptive immune response and, thus, they respond to immune checkpoint inhibitors more effectively [15].

By analysing GCs with lymphoid stroma (*n* = 24), Gullo et al. demonstrated that T cells are enriched at the invasive margin of GCs, as well as in EBV+ cases [16]. Derks et al. described heterogeneity in TIME among GC molecular subtypes. In their study, chromosomal instable GCs (*n* = 18) showed T cell exclusion and possessed CD8+ cells predominantly at the invasive margin [15]. We noticed a similar pattern in B7-H3-high GCs and, thus, propose that B7-H3 could be involved in this immune suppressive mechanism of TIME. Notably, the B7-H3 expression correlated with pT stage, indicating that the local immunosuppression is more pronounced in advanced cases of GC and, thus, might progress over time by expressing B7-H3 more intensively.

Although B7-H3 was formerly described as a co-stimulatory molecule of T cell responses [17], more and more current studies provide evidence of its co-inhibitory role [5,7]. Similar to other immune checkpoint molecules, the dual effect on immune system could indicate that B7-H3 has more than one binding partner. Hashiguchi et al. demonstrated that murine B7-H3 binds to TREM-like transcript 2 (TLT-2) receptor expressed on CD8+ T cells and, thus, causes their activation [18]. In humans, however, such interaction has not been found. It is noted that B7-H3 inhibits the activation and proliferation of CD4+ and CD8+ T cells, as well as the production of IFN-γ and IL-2 [19]. B7-H3 may regulate TCR-mediated gene transcription and induce T cell suppression by inhibiting or modulating the nuclear factor of activated T cells (NFAT), nuclear factor κ B (NF-κB) and activator protein-1 (AP-1) [20].

The association between B7-H3 positivity and CD8 expression has been reported. Guo et al. examined the expression of PD-L1, B7-H3 and B7-H4 during different stages of gastric carcinogenesis. In their set of 50 GCs, 78% cases were B7-H3 positive, and a negative correlation between B7-H3 in the tumour and the density of CD8-expressing cells was reported [9]. Here, we show similar results in a sample set of GCs of Western origin. Interestingly, in their study the B7-H3 was primarily expressed in the tumour/parenchymal cells and immune cells, but not in stromal cells. In the study of Zhan et al. (*n* = 268), B7-H3 expression in stromal cells was described. By examining the co-expression of B7-H3 with CD31 and alpha-SMA, they detected stromal B7-H3 expression in 62.7% of GC cases, mainly in cancer-associated fibroblasts, which were the major component of the tumour stroma [8]. This corresponds to our observation that B7-H3 appears to be expressed mainly in the stromal compartment of GC.

In conclusion, the results indicate that B7-H3 within GC stroma could also play a role in regulating cell-mediated immune responses against cancer in patients of Western descent. Thus, it could serve as a great target of interest for different novel cancer immunotherapeutic strategies, especially in immunologically “cold” tumours.

## 4. Materials and Methods

### 4.1. Study Population

A total of 98 gastric adenocarcinoma cases were selected by stratified sampling out of a consecutive series of GC patients treated with primary total or partial gastrectomy from the archive of the Institute of Pathology, University Hospital Schleswig-Holstein, Kiel, Germany. The following clinicopathological characteristics were collected: gender, age at diagnosis, tumour location, tumour phenotype by Lauren, pTNM stage, histological grade (G), UICC stage, as well as survival data. All patient-related data were pseudonymized. Two cases were excluded from the cohort as there was no clear invasive margin evaluable in the slides. Finally, a total of 96 cases (intestinal, *n* = 48; diffuse, *n* = 27; mixed, *n* = 6; unclassified, *n* = 15) were included in the study. Specimens were routinely fixed in formalin and embedded in paraffin.

### 4.2. Immunohistochemistry

Immunohistochemistry was performed automatically on whole tissue sections using a consecutive double staining protocol on a Leica BOND-MAX (Leica Biosystems, Wetzlar, Germany). In the first step, antigen retrieval was performed using BOND Epitope Retrieval Solution 2 (Leica Biosystems, Wetzlar, Germany) for 20 min, followed by incubation with anti-human B7-H3 (D9M2L, 1:50, Cell Signaling Technology, Leiden, Netherlands). Detection and visualization were done with the Bond Polymer Refine Detection kit (Leica Biosystems, Wetzlar, Germany). In the second staining step, antigen retrieval was performed again using BOND Epitope Retrieval Solution 2 (Leica Biosystems, Wetzlar, Germany), followed by the second antibody incubation with anti-human CD8α (4B11, 1:100, Leica Biosystems, Wetzlar, Germany). Finally, detection and visualization were done with the Bond Polymer Refine Red Detection (Leica Biosystems, Wetzlar, Germany).

### 4.3. Evaluation of B7-H3 Immunostaining

B7-H3 expression was scored manually by two pathologists (D.U. and C.R.) for staining intensity and localization. Four representative cases were selected for the adjustment of staining intensity (0, negative; 1, low; 2, moderate; 3, strong staining reaction) and used as reference standard for the evaluation of the entire cohort. Inconsistent results were resolved by consensus review. For statistical purposes, cases were divided into B7-H3-low (score 0/1) and B7-H3-high (score 2/3) groups as proposed by a previous study [8].

### 4.4. Digital Image Analysis

Digital images of tissue sections were obtained using a Leica SCN400 microscopic whole-slide scanner (Leica Microsystems, Wetzlar, Germany) at 40× nominal magnification, which corresponds to a resolution of 0.25 µm per pixel. To detect CD8+ T cells, digital image analysis with Definiens Tissue Studio (version 4.4.3, Definiens AG, Munich, Germany) was performed. The following detection settings were used: 20× nominal magnification for digital analysis; nucleus detection with haematoxylin threshold of 0.15 and typical nucleus size of 25 µm^2^; cell simulation with 2 µm of maximum growth as the optimal threshold for cell border generation; cell classification by double staining immunohistochemistry markers with red chromogen threshold of 0.32 and brown chromogen threshold of 3.0, respectively. These settings were used for all the images and carried out by a single author, who was blinded to patient outcomes. To avoid the false results due to different artefacts (folded tissue, holes, and large areas of necrotic debris and mucin pools), marking of tumour compartments was done manually by using the viewer and painting program VMP. The tumour centre (TC) was defined as intratumoural area comprising cancer cells and desmoplastic tumour stroma with no direct connection to the peritumoural non-tumourous tissue. The invasive front (IF) was defined as a narrow band-like area at the tumour–host interface with a width of up to 250 µm between the invasive margin of GC and adjacent non-tumourous tissue.

### 4.5. Statistical Analysis

Data were analysed using SPSS v25.0.0.2 (IBM Corporation, New York, NY, USA). Differences in clinicopathological variables between B7-H3 staining intensity groups were determined using Kendall’s tau test for ordinal variables and Fischer’s exact test for non-ordinal variables. Survival data were analysed by the Kaplan–Meier method and compared using log-rank test. Differences in CD8+ T cell densities between B7-H3 groups were analysed using Wilcoxon Signed Ranks test and Mann–Whitney U test. A significance level of *p* < 0.05 was assumed.

### 4.6. Bioinformatic Analysis

Data from TCGA STAD cohort were retrieved for quantitative validation analysis using TCGA Xena computational tool (UC Santa Cruz, http://xena.ucsc.edu (accessed on 14 February 2021)) [10]. The effect of B7-H3 (gene name: CD276) expression on CD8+ T cell infiltration and survival data in TCGA cohort was analysed by using TIMER2.0 web platform (http://timer.cistrome.org (accessed on 14 February 2021)) [11].

## Figures and Tables

**Figure 1 ijms-22-02129-f001:**
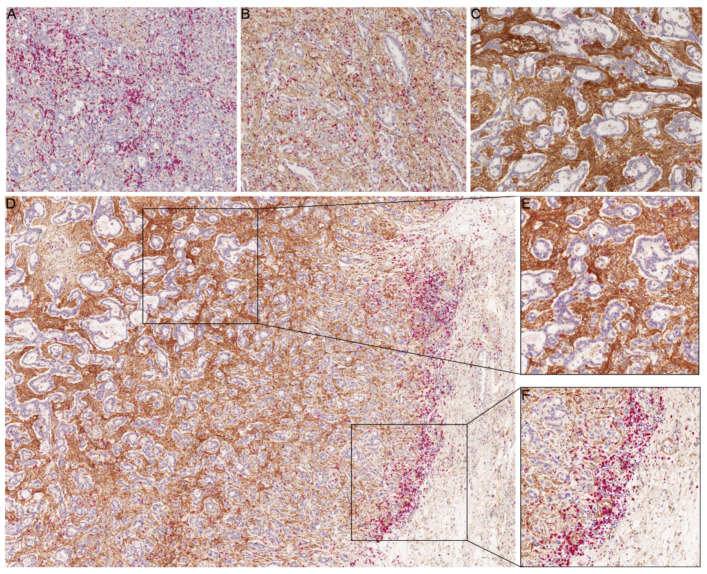
Immunohistochemical characterization of B7-H3 expression in gastric cancer in relation to CD8+ T cells. (**A**–**C**) Representative tissue sections of weak intensity (**A**), moderate intensity (**B**), and strong intensity (**C**) of B7-H3-immunostaining (brown labelling). The density of the CD8+ T cells (red labelling) in the tumour centre correlated inversely with the intensity of stromal B7-H3-immunostaining. (**D**–**F**) Distribution of CD8+ T cells (red) between tumour compartments in relation to B7-H3 expression (brown; **D**). While the tumour centre showed strong B7-H3 immunostaining with only few CD8+ T cells (**E**), CD8+ T cells were enriched at the invasion front of the stromal compartment (**F**). Anti-B7-H3- and anti-CD8-double-immunostaining; haematoxylin counterstain. Original magnifications 100-fold (**A**–**C**), 50-fold (**D**), and 200-fold (**E**,**F**).

**Figure 2 ijms-22-02129-f002:**
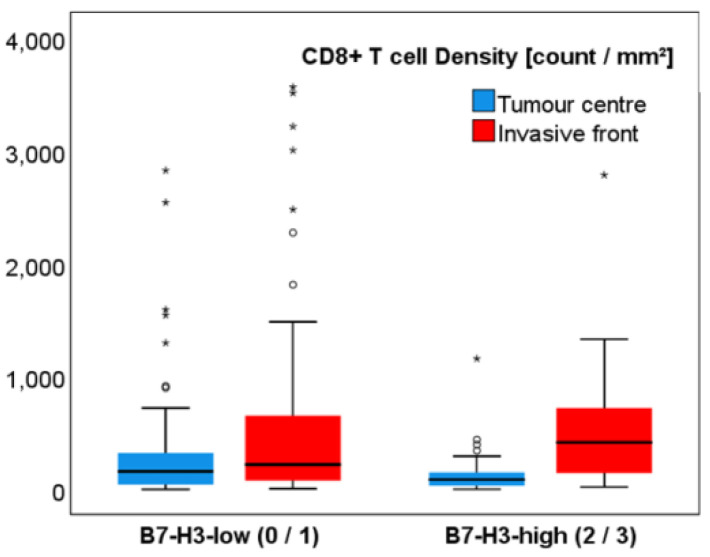
Boxplots of CD8+ T cell densities between tumour compartments in B7-H3-low and B7-H3-high groups. The median CD8+ T cell densities differed significantly between tumour centre and invasive front (*p* < 0.001), and tumours of B7-H3-high group showed greater spatial differences of CD8+ T cells when compared to the B7-H3-low group (*p* < 0.001); * extreme values.

**Table 1 ijms-22-02129-t001:** Clinicopathological patient characteristics and correlation with B7-H3 status.

Characteristics	Valid/Missing		B7-H3 Low		B7-H3 High		*p*-Value
	*n*	(%)	*n*	(%)	*n*	(%)	
Total	96	(100)	55	(57.3)	41	(42.7)	
Gender	96/0						0.061 ^2^
Male	57	(59.4)	28	(49.1)	29	(50.9)	
Female	39	(40.6)	27	(69.2)	12	(30.8)	
Age	96/0						0.410 ^1^
<68 years	50	(52.1)	31	(62.0)	19	(38.0)	
≥68 years	46	(47.9)	24	(52.2)	22	(47.8)	
Location	95/1						0.003 ^2^
Proximal stomach	37	(38.9)	14	(37.8)	23	(62.2)	
Distal stomach	58	(61.1)	40	(69.0)	18	(31.0)	
Laurén phenotype	96/0						<0.001 ^2^
Intestinal	48	(50.0)	17	(35.4)	31	(64.6)	
Diffuse	27	(28.1)	25	(92.6)	2	(7.4)	
Mixed	6	(6.3)	2	(33.3)	4	(66.7)	
Unclassified	15	(15.6)	11	(73.3)	4	(26.7)	
Grade	90/6						0.173 ^1^
G1/G2	17	(18.9)	7	(41.2)	10	(58.8)	
G3/G4	73	(81.1)	45	(61.6)	28	(38.4)	
pT category	96/0						0.013 ^1^
pT1/pT2	22	(22.9)	18	(81.8)	4	(18.2)	
pT3/pT4	74	(77.1)	37	(50.0)	37	(50.0)	
pN category	96/0						1.000 ^1^
pN0	32	(33.3)	18	(56.2)	14	(43.8)	
pN+	64	(66.7)	37	(57.8)	27	(42.2)	
M category	91/5						0.591 ^1^
M0	74	(81.3)	41	(55.4)	33	(44.6)	
M1	17	(18.7)	11	(64.7)	6	(35.3)	
UICC stage	91/5						0.778 ^1^
IA/IB	16	(17.6)	13	(81.2)	3	(18.8)	
IIA/IIB	25	(27.4)	10	(40.0)	15	(60.0)	
IIIA/IIIB/IIIC	33	(36.3)	18	(54.5)	15	(45.5)	
IV	17	(18.7)	11	(64.7)	6	(35.3)	
pR status	89/7						0.231 ^1^
pR0	82	(92.1)	45	(54.9)	37	(45.1)	
pR1/pR2	7	(7.9)	6	(85.7)	1	(14.3)	
Overall survival (months)							0.784 ^3^
Total/events/censored	89/64/25		51/38/19		38/26/12		
Median survival	17.5		17.5		16.6		
95% C.I.	10.7–24.3		9.0–26.0		10.3–22.9		
Cause-specific survival (months)							0.990 ^3^
Total/events/censored	82/50/32		47/28/19		35/22/13		
Median survival	18.4		17.5		18.4		
95% C.I.	9.6–27.3		6.6–28.5		12.8–24.1		

^1^ Kendall’s tau test, ^2^ Fisher’s exact test, ^3^ log-rank test; UICC, Union for International Cancer Control.

**Table 2 ijms-22-02129-t002:** The median density of CD8+ T cells between tumour compartments in relation to B7-H3 expression.

	*n*	CD8+ T Cell Density (cells/mm^2^)							
		Tumour Centre				Invasive Front			
		Q1	Median	Q3	Range	Q1	Median	Q3	Range
Total	96	33.1	112.5	214.9	0.0–2822.6	112.9	258.1	704.3	4.7–3560.1
B7-H3 expression									
B7-H3 low (0/1)	55	41.2	157.7	328.4	0.0–2822.6	65.2	218.7	658.3	4.7–3560.1
B7-H3 high (2/3)	41	30.4	86.4	149.9	0.9–1156.0	134.0	414.9	737.6	20.6–2781.4

Q1 and Q3 denote first and third quartile.

## Data Availability

The data presented in this study are available on request from the corresponding author. The TCGA STAD dataset is available in The Cancer Genome Atlas website (https://cancergenome.nih.gov/ (accessed on 14 February 2021)).

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
