# Peer review of "Gastric Carcinomas with Stromal B7-H3 Expression Have Lower Intratumoural CD8+ T Cell Density"

_ijms, 2021, doi:10.3390/ijms22042129_

Round 1
Reviewer 1 Report
In this manuscript, the authors analyzed that with the higher expression of B7-H3 in stromal cells, CD8+ T cells were suppressed in the tumor region, suggesting that stromal-B7-H3 is associated with tumor escape mechanisms in GCs. And they found that the survival demonstrates no difference between B7-H3-low- and high- groups. In general, the topic has some novelty and the description is accurate to support the conclusion. Several major concerns need to be addressed before drawing a solid conclusion:
1, As we know gastric cancer can be divided as lauren’s intestinal and diffuse type and these two subtypes are absolutely two different types. The authors need to analyze the stromal B7-H3 expression in these two subtypes respectively. Actually the diffuse type gastric cancer is often with more stromal infiltration.
2, How B7-H3 abundance is associated with low CD8+ cell intratumoral infiltration? By releasing some cytokines to quench CD8+ infiltration or blocking its migration? As from the peritumoral region, the CD8+ infiltration is more prominent than the B7-H3-low group. The authors need to address this crucial point as I remember a paper published by a Japanese group, they described the survival difference based on the infiltration of CD8+ infiltration in intra- or peri- tumoral region.
3, Any supportive dataset also demonstrates the same trend with your findings? Such as TCGA cohort. It is better to analyze the TCGA in terms of B7-13 expression/CD8 signature in intestinal and diffuse type gastric cancer respectively. I personally deduced you might get some findings that B7-H3 is highly expressed in diffuse type gastric cancer and associated with poor survival. It is only my assumption, might not be correct, but need data to support it.
Author Response
1, As we know gastric cancer can be divided as lauren’s intestinal and diffuse type and these two subtypes are absolutely two different types. The authors need to analyze the stromal B7-H3 expression in these two subtypes respectively. Actually the diffuse type gastric cancer is often with more stromal infiltration.
The association between B7-H3 stromal expression and gastric cancer phenotypes (including intestinal- and diffuse type) are summarized in Table 1. Briefly, the study included 48 intestinal- and 27 diffuse-type gastric carcinomas. In the intestinal-type subgroup, 64.6% showed moderate or strong staining and 35.4% - negative or low staining. Most of the diffuse-type gastric carcinomas (92.6%) showed no- or low expression.
2, How B7-H3 abundance is associated with low CD8+ cell intratumoral infiltration? By releasing some cytokines to quench CD8+ infiltration or blocking its migration? As from the peritumoral region, the CD8+ infiltration is more prominent than the B7-H3-low group. The authors need to address this crucial point as I remember a paper published by a Japanese group, they described the survival difference based on the infiltration of CD8+ infiltration in intra- or peri- tumoral region.
Thank you very much for raising this important point. The exact role of B7-H3 in regulating CD8+ could be dependent on the type of TIME, however this can be only speculated because its binding partner(s) is/are still unknown. We have added one paragraph to the manuscript which discusses the possible theories of B7-H3 and CD8+ T cell interaction.
Changes in manuscript: Discussion, lines 158-166.
“Similarly to other immune checkpoint molecules, the dual effect on immune system could indicate that B7-H3 has more than one binding partner. Hashiguchi et al. demonstrated that murine B7-H3 binds to TREM-like transcript 2 (TLT-2) receptor ex-pressed on CD8+ T cells and, thus, causes their activation [18]. In humans, however, such interaction has not been found. It is noted that B7-H3 inhibits both CD4 and CD8 T cell activation and proliferation, and production of IFN-γ and IL-2 [19]. B7-H3 may regulate TCR-mediated gene transcription and induce T cell suppression by inhibiting or modulating nuclear factor of activated T cells (NFAT), nuclear factor κ B (NF-κB) and activator protein-1 (AP-1) [20].”
3, Any supportive dataset also demonstrates the same trend with your findings? Such as TCGA cohort. It is better to analyze the TCGA in terms of B7-13 expression/CD8 signature in intestinal and diffuse type gastric cancer respectively. I personally deduced you might get some findings that B7-H3 is highly expressed in diffuse type gastric cancer and associated with poor survival. It is only my assumption, might not be correct, but need data to support it.
Thank you for this suggestion. In TCGA STAD cohort, the expression of B7-H3 was significantly and negatively correlated with CD8+ T cell infiltration. As in our cohort, no statistically significant difference was found in the survival between B7-H3 expression groups in the validation dataset. We have added new data of TCGA cohort in the Results.
Changes in manuscript: new paragraph added to Results, lines 114-124.
“2.4. TCGA dataset analysis
Data from The Cancer Genome Atlas Stomach Adenocarcinoma (TCGA STAD) cohort were retrieved for quantitative validation analysis using TCGA Xena computational tool (UC Santa Cruz) [10]. The B7-H3 mRNA expression in gastric cancer (TCGA STAD, n = 415) was significantly higher than in normal gastric tissue (TCGA STAD, n = 35) (p < 0.001, Wilcoxon test, Figure S1). The effect of B7-H3 expression on CD8+ T cell infiltration and survival data in TCGA cohort was analysed by using TIMER2.0 web platform [11]. In TCGA STAD cohort, the expression of B7-H3 was significantly and negatively correlated with CD8+ T cell infiltration (r = -0.224, p < 0.001, Figure S2). Similarly to our cohort, no statistically significant difference was found in OS between B7-H3 expression groups in the validation dataset (HR = 1.03, p = 0.661).”
Changes in manuscript: new paragraph added to Materials and Methods, lines 241-246.
“4.6. Bioinformatic analysis
Data from TCGA STAD cohort were retrieved for quantitative validation analysis using TCGA Xena computational tool (UC Santa Cruz, http://xena.ucsc.edu) [10]. The effect of B7-H3 (gene name: CD276) expression on CD8+ T cell infiltration and survival data in TCGA cohort was analysed by using TIMER2.0 web platform (http://timer.cistrome.org) [11].”
Changes in manuscript: Two supplementary figures are added, descriptions in lines 249-254:
“Supplementary materials: Figure S1: B7-H3 expression in gastric adenocarcinoma and normal gastric tissue. The B7-H3 mRNA expression in gastric cancer (TCGA STAD, n = 415) was signifi-cantly higher than in normal gastric tissue (TCGA STAD, n = 35) (p < 0.001, Welch's t-test, TCGA Xena computational tool). Figure S2: Relevance of B7-H3 expression to CD8+T cell infiltration. In TCGA STAD cohort, the expression of B7-H3 was significantly and negatively correlated with CD8+ T cell infiltration (r = -0.224, p < 0.001, TIMER2.0).”
Changes in manuscript: the information of TCGA dataset has been added to the Data Availability Statement, lines 268-269.
Reviewer 2 Report
The study describes the spatial and semi-quantitative expression of B7-H3 molecules on GC tumor in comparison with the presence of CD8T-cells in the same tissue samples.
The study is of interest since contributes to better decipher the role of the complex immune response in GC and with potential repercussions in the immune-checkpoint treatment response.
comments
Results: the number of cases (point 2.1) did not correspond to the 96 cases (60+13 =93!!)
Differences in tumors (n=60) and TIME (n=13) could be better described and discussed.
Table 1: "tumor specific survival" is not a correct title please change the terms: e.g recurrence, events
Kendall's test for ordinal variables and Fischer tests are not referred to adequately in table 1
Fig 1 C...the enlargement of the figure could be the same that in A and B.
TC and IF abbreviations could be specified when reported for the first time in the text
Author Response
- Results: the number of cases (point 2.1) did not correspond to the 96 cases (60+13 =93!!)
A total of 96 gastric cancer cases were included in this study, and 73 cases were B7-H3 positive. To avoid further misunderstandings, the following information was added to the manuscript:
Changes in manuscript: Results, added in lines 62-64
“A total of 96 cases (intestinal, n = 48; diffuse, n = 27; mixed, n = 6; unclassified, n = 15) were included in the study. Among 73 B7-H3 positive GCs (76%), B7-H3 expression was found within the tumour stroma (n = 60) or both in stromal cells and neoplastic cells (n = 13).”
- Differences in tumors (n=60) and TIME (n=13) could be better described and discussed.
We agree with the reviewer that it would be interesting to compare both cohorts (B7-H3 stromal positivity vs. B7-H3 positivity in tumour cells). However, as already mentioned in lines 67-69, it was very small (n =13), therefore we focused on cases with B7-H3 stromal expression (n = 60). Although we hope that these findings will be helpful in the design of future studies showing another, possibly relevant pattern of B7-H3 expression in gastric cancer. Thus, larger studies in this area should be performed.
- Table 1: "tumor specific survival" is not a correct title please change the terms: e.g recurrence, events
Thank you for this valuable comment. The term “tumour specific survival” was changed to “cause-specific survival” to correspond to the definition of the National Cancer Institute (NCI) Dictionary of Cancer Terms: The length of time from either the date of diagnosis or the start of treatment for a disease, such as cancer, to the date of death from the disease. Patients who die from causes unrelated to the disease are not counted in this measurement. (www.cancer.gov/publications/dictionaries/cancer-terms/def/cause-specific-survival).
Changes in manuscript: Term “tumour specific survival” was changed to “cause-specific survival” in Table 1 and line 91; the relevant abbreviations were changed from TSS to CSS, respectively (lines 91-92).
- Kendall's test for ordinal variables and Fischer tests are not referred to adequately in table 1
Thank you for this comment. The relevant tests have been added to the clinicopathological variables.
Changes in manuscript: Table 1: in rows Gender and Location, Fisher’s exact test (marked with 2 ) is used instead of Kendall’s tau test (marked with 1 )
- Fig 1 C...the enlargement of the figure could be the same that in A and B.
The enlargement of Figure 1C corresponds to Figures 1A and 1B.
- TC and IF abbreviations could be specified when reported for the first time in the text.
Changes in manuscript: The explanations of abbreviations “tumour centre (TC)” and “invasive front (IF)” were added to the text in line 97.
Round 2
Reviewer 1 Report
The authors have successfully addressed all my concerns. No other issues raised.